# Natural Occurrence of *Alternaria* Toxins in Citrus-Based Products Collected from China in 2021

**DOI:** 10.3390/toxins15050325

**Published:** 2023-05-09

**Authors:** Xiaomin Han, Wenjing Xu, Luxinyi Wang, Ruina Zhang, Jin Ye, Jing Zhang, Jin Xu, Yu Wu

**Affiliations:** 1Key Laboratory of Food Safety Risk Assessment, Ministry of Health, China National Center for Food Safety Risk Assessment, Beijing 100021, China; 2College of Biochemical Engineering, Beijing Union University, Beijing 100101, China; 3Department of Dermatology, Beijing Friendship Hospital Capital Medical University, Beijing 100050, China; 4Academy of National Food and Strategic Reserves Administration, Beijing 102600, China

**Keywords:** *Alternaria* toxins, citrus-based products, China, UPLC-ESI-MS

## Abstract

A total of 181 citrus-based products, including dried fruits, canned fruits, and fruit juices, collected from China and from abroad in 2021 were analyzed for the four *Alternaria* toxins (ALTs): alternariol (AOH), alternariol monomethyl ether (AME), tentoxin (TEN), and tenuazonic acid (TeA) via ultra-high-performance liquid chromatography–electrospray ionization–tandem mass spectrometry (UPLC-ESI-MS). Although the concentrations of the four ALTs varied by product and geographically, TeA was the predominant toxin followed by AOH, AME, and TEN. Products made in China showed higher levels of ALTs than those made abroad. Maximum levels of TeA, AOH, and AME in analyzed domestic samples were 4.9-fold, 1.3-fold, and 1.2-fold, respectively, higher than those in imported products. Furthermore, 83.4% (151/181) of the analyzed citrus-based products were contaminated with at least two or more ALTs. There were significant positive correlations between AOH and AME, AME and TeA, and TeA and TEN in all analyzed samples. More importantly, the solid and the condensed liquid products had higher concentrations of ALTs than the semi-solid product samples, as well as tangerines, pummelos, and grapefruits compared to the other kinds of citrus-based products. In conclusion, co-contamination with ALTs in commercially available Chinese citrus-based products was universal. Extensive and systematic surveillance of ALTs in citrus-based products, both domestic and imported, is required to obtain more scientific data for the determination of the maximum allowable concentrations of ALTs in China.

## 1. Introduction

*Alternaria* toxins (ALTs) are a series of mycotoxins produced by *Alternaria* species, which are harmful to human and animal health [1,2]. Some ALTs induce fetotoxic and teratogenic effects in animals. Individual toxins such as AOH or AME are mutagenic and clastogenic in vitro. Due to their harmful effects, ALTs cause concern for public health. According to the European Food Safety Authority (EFSA) report, more than 70 ALTs have been identified and separated into five categories according to their chemical structure [2]. Alternariol (AOH), alternariol monomethyl ether (AME), tentoxin (TEN), and tenuazonic acid (TeA) are the most common ALTs, and their chemical structures are shown in Figure 1. ALTs can induce cytotoxic, fetotoxic, and teratogenic effects in animals. Toxins such as AOH or AME are mutagenic and genotoxic in many in vitro systems. Epidemiological studies have found that certain ALTs in grains might be responsible for esophageal cancer in China [3]. We can find them in food, such as cereals and cereal-based products, tomato sauces, fig wines, and sunflower seeds [2,4]. There are limited reports of natural occurrence in fruit.

The genus *Citrus* belongs to the angiosperm subfamily *Aurantioideae* of the *Rutaceae* family, and it is the second-largest cash fruit crop, covering lemon, orange, tangerine, pummelo, and grapefruit [5]. The annual yield of citrus is high in China and reached 49 million tons in 2021 [6]. Citrus and its products, including juice, canned fruits, and dried products, have a large consumption market. According to the statistics, the average per capita citrus consumption reached 20.9 kg in 2017. People use citrus for eating, processing, and medicine [5,7]. So, the food safety of citrus-based products is important for public health. Citrus *Alternaria* brown spot usually infects citrus, and *Alternaria* species produce several ALTs in citrus pre- and post-harvest [8]. Furthermore, ALTs have attracted great interest from the global scientific community due to their toxicity and high occurrences in many countries, such as South Africa, Turkey, Iran, Italy, the Czech Republic, Germany, and Brazil [2,4,9,10,11,12]. While the vast majority of reports on the natural occurrence of ALTs in China focus on grain, feed, and related products, limited reports are available on fruit and fruit-based products [13,14,15,16,17]. However, Shi et al. reported that citrus from Southwest China is becoming more vulnerable to infection by *Alternaria* species and contamination by ALTs [18,19]. Due to global climate warming and the increase in imported products, *Alternaria* species may become the primary pathogens from the auxiliary pathogen group, causing considerable losses of crop yield and economic income and more contamination by ALTs. In the year 2021, the COVID-19 pandemic left great impact on all walks of life. The production and consumption of citrus-based products was also affected. So, it is necessary to figure out the food quality of citrus-based products and its potential risk.

Due to limited reports about the natural occurrence and toxicity, no countries or organizations have published official maximum levels (MLs) for the four ALTs. The CONTAM panel of EFSA in the food chain applied the threshold of toxicological concern (TTC) approach to the four ALTs in 2011 [2]. EFSA also recommended the collection of more data on the natural occurrences of ALTs in fruit [2]. The purpose of this paper is to elucidate the contamination levels in citrus-based products collected from China of the four main ALTs: AOH, AME, TeA, and TEN. The results obtained in this study will provide a scientific basis for the setting of MLs for ALTs in China and their dietary exposure risk assessment from the consumption of citrus-based products.

## 2. Results

### 2.1. Natural Occurrence of the Four ALTs in Commercially Available Chinese Citrus-Based Products in 2021

The contamination levels of the four ALTs in 181 commercially available Chinese citrus-based products are presented in Table 1. TeA was the predominant mycotoxin in both frequency and concentration. Some 78.3% (142/181) of samples were positive for TeA with an average level of 17.8 μg/kg (range: 13.7 μg/kg to 312.7 μg/kg, median = 14.3 μg/kg). Among all samples analyzed, 1 sample contained TeA at a level higher than 300 μg/kg, 2 samples had levels between 50 μg/kg and 100 μg/kg, and 178 samples had levels below 50 μg/kg. Secondly, AOH had a positive rate of 60.8% (110/181) (average = 7.8 μg/kg, range = 1.1 μg/kg to 62.4 μg/kg, median = 6.5 μg/kg). There was one sample with AOH at a level higher than 50 μg/kg. For AME, 128 out of 181 (70.7%) samples had detectable concentrations ranging from 0.7 μg/kg to 13.4 μg/kg (mean = 1.2 μg/kg, median = 0.8 μg/kg). TEN showed the lowest frequency, with the lowest average concentration and maximum (positive = 33.7%, average = 0.9 μg/kg, maximum = 2.9 μg/kg). In addition, no sample in this current study had AME and TEN concentrations higher than 50 μg/kg. Therefore, the contamination trends of the four ALTs, from high to low, followed the order: TeA > AOH > AME > TEN. This finding is different from the results from Hickert et al. [9], Xu et al. [14], and Jiang [19]. This might be due to the difference between sample matrices of multiple varieties.

### 2.2. Contamination and Co-Contamination with the Four ALTs in Commercially Available Chinese Citrus-Based Products in 2021

From Table 2 and Figure 2, it can be concluded that the contamination and co-contamination with the four ALTs commercially available Chinese citrus-based products were frequent. Importantly, 151 samples (83.4%, 151/181) were contaminated with at least two mycotoxins. However, only 12 (6.6%, 12/181) and 18 (10.0%, 18/181) samples had no toxins or only one detected toxin, respectively. Samples with three toxins had the highest frequency (43.6%, 79/181), followed by those with two toxins (28.2%, 51/181) and four toxins (11.6%, 21/181). For the 79 samples co-contaminated by three toxins, the combination of AOH-AME-TeA was detected most frequently in 50 (27.6%, 50/181) samples, followed by AME-TeA-TEN, AOH-AME-TeN, and AOH-TeA-TEN, with a frequency of 8.3% (15/181), 3.9% (7/181), and 3.9% (7/181), respectively. In terms of co-contamination by two toxins, there were 20 (11.0%, 20/181) samples positive for AME and TeA, 13 (7.2%, 13/181) samples positive for AOH and TeA, 7 (3.9%, 7/181) samples positive for AOH and AME, 5 (2.8%, 5/181) samples positive for TeA and TEN, 4 (2.2%, 4/181) samples positive for AME and TEN, and 2 (1.1%, 2/181) samples positive for AOH and TEN. For the 18 samples contaminated by only one toxin, there were 10 (5.5%, 10/181), 5 (2.8%, 5/181), and 3 (1.7%, 3/181) samples contaminated by TeA, AME, and AOH, respectively. Moreover, a significant linear correlation in concentrations was found between AOH and AME (*r* = 0.448, *p* < 0.01), AME and TeA (*r* = 0.291, *p* < 0.01), and TeA and TEN (*r* = 0.322, *p* < 0.05) in all detected citrus-based samples, as shown in Figure 3. These results are similar to those previously reported for wheat-based products by Zhao et al. [13].

### 2.3. Domestic and Imported Distribution of the Four ALTs in Commercially Available Chinese Citrus-Based Products in 2021

The natural occurrences of the four ALTs for samples from products made domestically or imported are given in Table 3 and Figure 4. The contamination with citrus-based products by the four ALTs varied geographically. Higher concentrations of the four toxins in terms of either the average, median, or maximum were all found in samples made in China. The levels of natural contaminations of TeA in both domestic and imported samples were significantly higher than those of the other three toxins (Kruskal–Wallis test, *p* < 0.05). TeA had the highest positive rate (76.5% for domestic samples and 81.0% for imported samples) and the highest average concentration (mean = 19.0 μg/kg, median = 14.2 μg/kg, and range = 13.7 μg/kg to 312.7 μg/kg for domestic samples, mean = 16.3 μg/kg, median = 14.4 μg/kg, range = 13.7 μg/kg to 53.2 μg/kg for imported samples). More importantly, the maximum level of TeA in domestic samples was 4.9 times higher than that in imported samples. AOH was second with a positive rate of 60.8% (62/102) and 59.5% (47/79) and an average of 8.3 μg/kg (median: 6.6 μg/kg, range: 1.1 μg/kg to 62.4 μg/kg) and 7.2 μg/kg (median: 6.0 μg/kg, range: 1.1 μg/kg to 27.7 μg/kg) in domestic and imported samples, respectively. Additionally, AME was detected in 73.5% (75/102) of domestic samples and 67.1% (53/79) of imported samples with an average level of 1.3 μg/kg (median = 0.8 μg/kg, range = 0.7 μg/kg to 14.4 μg/kg) for domestic products and 1.0 μg/kg (median = 0.8 μg/kg, range = 0.7 μg/kg to 6.6 μg/kg) for imported products, respectively. TEN had the lowest positive rate and the lowest average concentration (31.4% and 0.9 μg/kg for domestic samples and 36.8% and 0.9 μg/kg for imported samples). A significant linear correlation in concentrations was observed between TeA and AME (*r* = 0.265, *p* < 0.05) and AOH and AME (*r* = 0.406, *p* < 0.01) in domestic samples. There was also a significant linear correlation in concentrations between AOH and AME (*r* = 0.564, *p* < 0.01), TeA and AME (*r* = 0.442, *p* < 0.01), and AME and TEN (*r* = 0.478, *p* < 0.05) in imported samples.

In short, we conclude that the samples from Chinese sources contained higher amounts of ALTs than those from other countries. The distributions of the mean and median concentrations of the four toxins in domestic and imported samples followed the same order, from high to low: TeA > AOH > AME > TEN.

### 2.4. Differences in the Physical Distribution of the Four ALTs in Commercially Available Chinese Citrus-Based Products in 2021

The differences in the natural distribution of the four ALTs in solid (dried fruits and preserved fruits), semi-solid (canned fruits and jams), and liquid (fruit juices) samples are shown in Table 4. Firstly, the highest contamination incidences, medians, and maximums of TeA, AOH, AME, and TEN were found in solid or liquid samples compared with the semi-solid samples. The highest incidences of TeA, AOH, AME, and TEN were found in the three physical-form samples, at 93.3% (14/15), 80% (12/15), 86.7% (13/15), and 36.4% (51/140), respectively. Importantly, the positive rate of the four ALTs in the three types of detected samples followed the same order, TeA > AME > AOH > TEN, while this order was distinct for the trends in the average and median: TeA > AOH > AME > TEN.

In terms of the 15 solid samples, the maximum and the average levels of the four ALTs were 23.9 μg/kg and 12.8 μg/kg (AOH), 13.4 μg/kg and 2.9 μg/kg (AME), 18.2 μg/kg and 14.7 μg/kg (TeA), and 0.8 μg/kg and 0.8 μg/kg (TEN), respectively. The highest values for AOH (24.0 μg/kg) and TeA (18.2 μg/kg) were found in one dried tangerine peel sample. Among the 26 semi-solids, the average, median, and maximum of the four ALTs in analyzed samples followed the order, from high to low, TeA > AOH > AME > TEN, while their concentrations were lower than those detected in solid samples. In the 140 liquid samples, the prevalence, average, median, and maximum values of the four ALTs followed the same trend as the semi-solid samples, except for the positive rate of AME. Furthermore, the maximum levels of AOH, TeA, and TEN in the 181 analyzed samples were all found in liquid samples (AOH: 62.4 μg/kg, TeA: 312.7 μg/kg, and TEN: 2.9 μg/kg), and the highest level of AME was found in 1 solid sample (AME: 13.4 μg/kg). Importantly, the liquid sample with the highest level of TeA was a concentrated pummelo juice, which also had the highest TEN (2.9 μg/kg) value. In addition, a significant linear correlation in concentrations was found between AOH and AME (*r* = 0.451, *p* < 0.01), TeA and AME (*r* = 0.384, *p* < 0.01), and TeA and TEN (*r* = 0.321, *p* < 0.05) in liquid samples, and AOH and AME (*r* = 0.787, *p* < 0.05) in semi-solid samples. Therefore, solid and concentrated liquid samples were more likely to be contaminated with the four ALTs than other samples.

### 2.5. Varietal Distribution of the Four ALTs in Commercially Available Chinese Citrus-Based Products in 2021

To the best of our knowledge, orange-, tangerine-, lemon-, pummelo-, and grapefruit-based products are the most common citrus-based products. We can infer the natural distributions of the four ALTs in the four kinds of samples from Table 5. TeA was the predominant toxin in terms of frequency and concentration in the four kinds of samples. Although the concentrations of the four ALTs in the analyzed samples varied in different kinds of citrus (Kruskal–Wallis test, *p* < 0.05), the average and median of the four ALTs in different samples ranked from high to low followed the same: TeA> AOH > AME > TEN. All the above data were similar for mycotoxin occurrence in dried fruits from China [16]. For 100 orange samples, only 1 orange sample contained TeA at a level higher than 30 μg/kg (maximum = 36.0 μg/kg), and the maximum levels of the other three ALTs were all below 30 μg/kg.

For the 33 tangerine samples, the trends of the average and median concentration of the four ALTs were in line with the natural occurrence of ALTs in the orange samples, while the average and positive levels of TeA, AOH, and AME in tangerines were higher than those in oranges. There was no sample with levels of the four ALTs higher than 30 μg/kg (maximum = 23.9 μg/kg for AOH). Of the 31 lemon samples, only 1 sample had a TeA concentration higher than 30 μg/kg. The average and median levels of TeA and AOH in lemons were higher than those of AME and TEN. For the 17 pummelos and grapefruit samples, TeA was highest in the four kinds of citrus products by average (average = 37.3 μg/kg) and maximum (maximum = 312.7 μg/kg), and the maximum level of 312.7 μg/kg was found in 1 condensed grapefruit juice sample. A significant linear correlation in concentrations was found between AME and TEN (*r* = 1.000, *p* < 0.01) in orange samples, AOH and AME (*r* = 0.428, *p* < 0.01), and TeA and AME (*r* = 0.517, *p* < 0.01) in tangerine samples, and AME and TEN (*r* = 0.747, *p* < 0.05) in other samples, and no significant correlations were observed. In general, tangerine, pummelo, and grapefruit samples, especially those in dried or concentrated forms, were more contaminated with the four ALTs [20,21,22,23,24].

As far as we know, orange-, tangerine-, lemon-, pummelo-, and grapefruit-based products are the most common citrus products. We can infer the natural distributions of the four ALTs in the four kinds of samples from Table 5. TeA was the predominant toxin in terms of frequency and concentration in the four kinds of samples. Concentrations of the four ALTs in the analyzed samples varied by citrus variety (Kruskal–Wallis test, *p* < 0.05), but the average and median concentrations of the four ALTs in different samples, ranked from high to low, were the same: TeA> AOH > AME > TEN. For 100 orange samples, only 1 orange sample had a level of TeA higher than 30 μg/kg (maximum = 36.0 μg/kg), and the maximum levels of the other three ALTs were all below 30 μg/kg. Concerning 33 tangerine samples, the trends of the average and median concentration of the four ALTs were in line with the natural occurrence of ALTs in 100 analyzed oranges, while the average and positive levels of TeA, AOH, and AME in tangerines were higher than those in orange. There was no sample with concentrations of higher than 30 μg/kg of the four ALTs (maximum = 23.9 μg/kg for AOH). Regarding 31 lemon samples, only 1 product had a concentration of TeA higher than 30 μg/kg. The average and median levels of TeA and AOH in the analyzed lemons were higher than those of AME and TEN. For the 17 pummelos and grapefruit samples, TeA was highest in the four kinds of citrus products by average (average = 37.3 μg/kg) and maximum (maximum = 312.7 μg/kg), and the maximum of 312.7 μg/kg was from 1 grapefruit condensed juice sample. A significant linear correlation in concentrations was found between AME and TEN (*r* = 1.000, *p* < 0.01) in orange samples, AOH and AME (*r* = 0.428, *p* < 0.01), and TeA and AME (*r* = 0.517, *p* < 0.01) in tangerine samples, and AME and TEN (*r* = 0.747, *p* < 0.05) in others, and no significant correlations were found for pummelo and grapefruit samples.

## 3. Discussion

Due to the reduction in the planting area and output of apples in 2018, the yield of citrus has exceeded the apple output for the first time, and citrus became one of the most important fruits in China with the largest planting area and highest production [25,26]. Additionally, China ranks highly in terms of its production and consumption of citrus-based products. In China, there is a strong demand for citrus and citrus-based products [27,28,29]. Therefore, it is necessary to ensure the safety of citrus and citrus-based products in China. Citrus-based products are usually derived from oranges, pummelos, grapefruits, and lemons [21]. All of these fruits are easily contaminated with ALTs and *Alternaria* species.

Given the limited data on the natural occurrence of ALTs in food and their toxicity to humans and animals, there are no regulations for ALTs in food in China or abroad. However, the CONTAM panel from EFSA applied the TTC approach to ALTs, indicating that ALTs could cause harm to health and should be given more attention [2]. The European Commission has requested EFSA’s opinion on their risk to human and animal health and for more data on the presence of ALTs in food [2]. As far as we know, this is the first systematic report on the natural occurrence of ALTs in commercially available citrus-based products from China.

One important finding was that the contamination and co-contamination with the four ALTs in the analyzed citrus-based products were common, as similarly mentioned in the reports of Xu et al. and Zhao et al. [13,14,17]. Out of the total number of samples, 83.4% (151/181) of samples were contaminated with at least two ALTs, covering 79 samples (43.6%, 79/181) and 21 samples (11.6%, 21/181) with three and four ALTs simultaneously, respectively. Reasons for this might be that ALT-producing fungi such as *A. alternata* and *A. tenuissima* are widespread in agricultural plants and their surrounding natural environments, such as the soil and air [30,31]. *Alternaria* species can infect citrus and related products during planting, production, and processing [28]. The second important finding is that levels of ALTs in concentrated juices and solid samples were higher than in ordinary juice and semi-solid samples. Additionally, similar results for mycotoxin occurrence in Chinese dried fruits were obtained [16]. This phenomenon may be due water removal, drying, or heating during processing, resulting in higher concentrations of ALTs in the final product. Moreover, masked ALTs, such as alternariol-3-sulfate, alternariol-9-sulfate, alternariol-3-sulfate-9-glucoside, and alternariol-9-sulfate-3-glucoside, may be converted into the detected forms during processing [32,33]. Additionally, some moldy fruit may have been included during processing [28].

The third important finding is that the levels of the four ALTs in the analyzed positive samples made in China were higher than those made abroad. The reason for this is that the processing technique for citrus-based products such as juices, canned fruits, and dried products made in China is different from those practiced abroad and more lax. This may indicate that the processing methods and controlling measures for citrus-based products in China need to be improved [29]. Mechanized equipment and intelligent management should be further developed. In addition, the selection of raw materials and quality control need to be standardized and strict in China. The whole processing chain should be controlled under the Hazard Analysis Critical Control Point (HACCP) System with a combination of manual and automatized efforts. In addition, the results obtained in our study are in line with the trends seen for tomato- and citrus-based products reported by Zhao et al. [17]. The maximum levels of TeA, AOH, and AME in analyzed domestic samples in this current study were 4.9-fold, 1.3-fold, and 1.2-fold, respectively, higher than those in analyzed imported samples. However, the contamination trends of the four ALTs in this study were different from those in a previous report [17]. In addition, our findings demonstrate that the average concentrations of AOH, AME, TeA, and TEN in all analyzed positive samples were lower than those in wheat and wheat kernel samples [14,15].

The fourth important finding is that tangerines, pummelos, and grapefruits, especially for canned goods, dried, or concentrated products, were more contaminated with the four ALTs. The production of ALTs is always associated with infection by *Alternaria* species, and different species of *Alternaria* can invade different plants and produce different kinds of toxins due to genetic and environmental differences [1,2,20,21,22,23,24]. For example, *Alternaria alternate* is more likely to infect pummelos and grapefruits, which can produce TeA, AOH, and AME [1,2,20,21,22,23,24], and could thus explain our results.

The study demonstrated that contamination and co-contamination with the four ALTs in citrus-based products collected in Chinese from commercial markets are common. This finding raises concern about the risk associated with the consumption of citrus-based products in China. Thus, we should conduct more natural occurrence surveys to collect data to estimate dietary exposure to ALTs in Chinese populations. Furthermore, we should implement a combined and integrated risk assessment of contamination with the four ALTs in Chinese citrus-based samples. We also need to establish a national standard determination method to conduct systematic and national surveillance protocols.

## 4. Materials and Methods

### 4.1. Chemicals and Reagents

Standards for AOH, AME, TeA, and TEN (purity > 98%) were purchased from Qingdao Pribolab Biotech Co., Ltd. (Qingdao, China). Methanol and acetonitrile (LC-MS grade) were obtained from Fisher Scientific (Fair Lawn, NJ, USA). Ammonium bicarbonate (LC-MS grade) was purchased from Sigma-Aldrich (St. Louis, MO, USA). Phosphoric acid and sodium dihydrogen phosphate were both analytical-grade (Sinopharm Chemical Reagent Co., Ltd., Shanghai, China). Water was prepared using a Millipore Milli-Q System (Millipore, Bedford, MA, USA) with a conductivity ≥18.2 MΩ.cm at 25 °C.

### 4.2. Sample Collection and Preparation

A total of 181 citrus-based samples were collected from a supermarket in Beijing or purchased from the internet according to the annual output and market share, covering major brands of products all over the world. This included 100 oranges, 33 tangerines, 31 lemons, and 17 pummelos and grapefruits. Among them, 102 samples and 79 samples were from China and abroad, respectively. The imported samples came from 10 countries, including the US, Brazil, Argentina, Spain, Italy, the Czech Republic, Thailand, South Korea, Japan, and Malaysia. For liquid samples (fruit juices), each one was shaken and mixed sufficiently. Solid (dried fruits and preserved fruits) and semi-solid (canned fruits and jams) samples were mixed thoroughly and ground to a 20-mesh power using a Blender 8010ES (Warning Commercial, Stamford, CT, USA) and a sifter. Then, prepared products were kept at −20 °C prior to analysis.

### 4.3. Toxin Analysis

All collected samples were analyzed for TeA, TEN, AOH, and AME based on previous methods [13] with modifications. Briefly, a prepared sample (5 g) was mixed with 25 mL of acetonitrile, 0.05 mol/L sodium dihydrogen phosphate (pH 3.0), and methanol (45:45:10, *v*/*v*/*v*), and sonicated for 60 min before centrifugation at 10,000 r/min 10 min. The supernatant was transferred to a new 50 mL tube and diluted with water to 30 mL. A total of 6 mL of the diluted supernatant was mixed with another 15 mL of sodium dihydrogen phosphate (0.05 mol/L, pH 3.0) and further purified. The HLB solid-phase extraction (SPE) cartridge (Waters, Switzerland) was eluted with 5 mL of methanol followed by 5 mL of water. Secondly, the diluted extraction supernatant was loaded on the SPE cartridge and eluted with 5 mL of methanol followed by 5 mL of acetonitrile. The eluents were combined, dried, and reconstituted in 2 mL of methanol–water (10:90, *v*/*v*). Following centrifugation at 12,000 r/min 4 °C for 10 min, 1 µL of the extract was analyzed for the four ALTs [13]. An *Alternaria* toxin-free citrus-based sample was used as a base for the matrix-matched calibration standards for quantification.

### 4.4. UPLC Conditions

A UPLC-ESI-MS/MS system equipped with an Exion LC (Shimadzu, Kyoto, Japan) and a QTRAP^TM^ 5500 MS/MS system (AB Sciex, Foster City, CA, USA) were used to quantify the four ALTs simultaneously. MultiQuant^TM^ Version 3.0.2 software (AB Sciex, Foster City, CA, USA) was used for data analysis. Chromatographic separation was achieved using a C_18_ column (2.1 mm × 50 mm, 1.7 μm bead diameter, Waters, Milford, MA, USA). Temperature of the column and autosampler was set at 40 °C and 10 °C, respectively. A binary gradient with mobile phase A (1.0 mmol/L ammonium bicarbonate) and B (methanol) was programmed, and the flow rate was 0.3 mL/min.

### 4.5. MS/MS Conditions

The experiments were carried out in the negative electrospray ionization (ESI^−^) mode with an optimized dwell time of 200 ms. The curtain gas, the collision gas, Gas 1, and Gas 2 were set to 35 psi, medium, 60 psi, and 50 psi, respectively. The source desolvation temperature and the ion spray voltage were 500 °C and −5500 V, respectively. The parent ions (*m*/*z*) for AOH, AME, TeA, and TEN are 257.0, 271.0, 196.0, and 413.2, respectively. The quantitative and qualitative daughter ions are 213.0 and 147.1 for AOH, 256.0 and 228.0 for AME, 139.1 and 112.1 for TeA, and 141.1 and 271.1 for TEN, respectively.

### 4.6. Method Validation

The four ALT-free citrus-based samples were spiked with the four ALTs at three concentrations ranging from 2 mg/kg to 100 mg/kg for TeA and AOH, and 0.4 mg/kg to 20 mg/kg for AME and TEN, respectively. Each concentration was spiked in six samples and extracted and analyzed for the ALT recovery analysis. The recovery for the four ALTs ranged from 80% to 120%, and the interlaboratory RSDr calculation was below 10%. All of the above data indicate that this method is suitable for ALT detection in citrus-based samples.

### 4.7. Data Analysis

Significant differences in the concentrations of the four ALTs were found using the Kruskal–Wallis H test. Relationships of concentrations between any of the four ALTs were tested with Spearman correlation. All statistical analyses were carried out using SPSS software (version 20.0, IBM, Chicago, IL, USA).

## Figures and Tables

**Figure 1 toxins-15-00325-f001:**
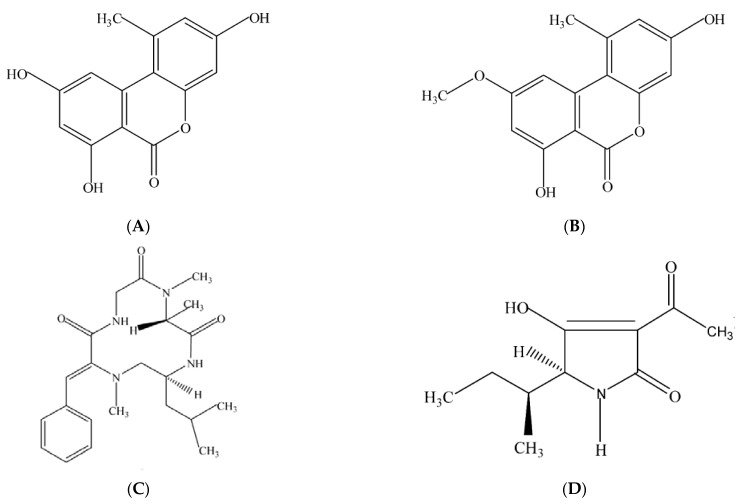
Chemical structures of (**A**) alternariol; (**B**) alternariol monomethyl ether; (**C**) tentoxin; and (**D**) tenuazonic acid.

**Figure 2 toxins-15-00325-f002:**
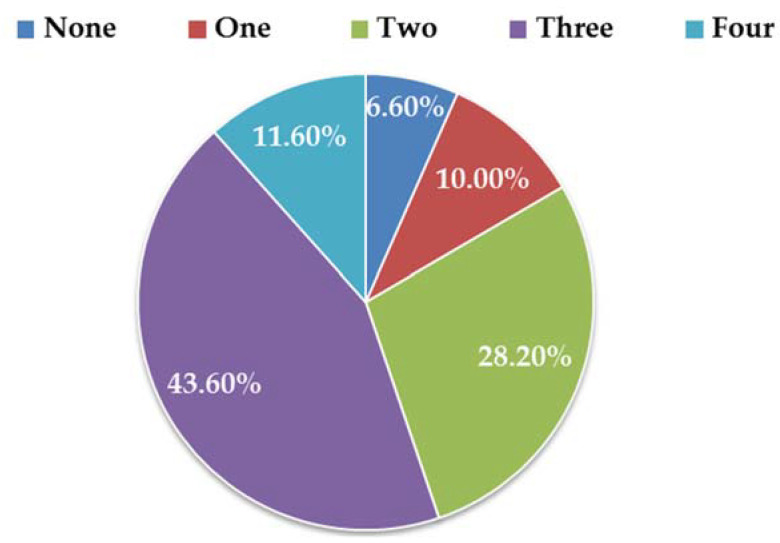
The contamination and co-contamination with the four ALTs in commercially available Chinese citrus-based products in 2021.

**Figure 3 toxins-15-00325-f003:**
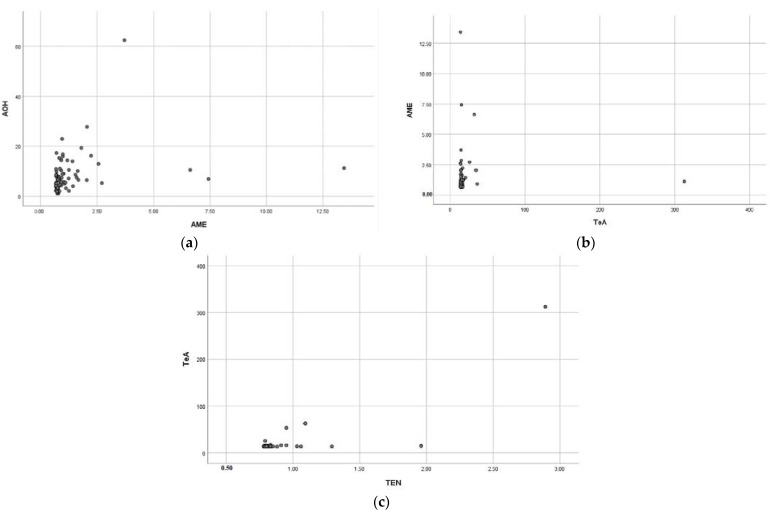
Correlation in concentrations of the four ALTs commercially available Chinese citrus-based products in 2021. (**a**) AOH vs. AME (*r* = 0.448, *p* < 0.01), (**b**) AME vs. TeA (*r* = 0.291, *p* < 0.01), (**c**) TeA vs. TEN (*r* = 0.332, *p* < 0.05).

**Figure 4 toxins-15-00325-f004:**
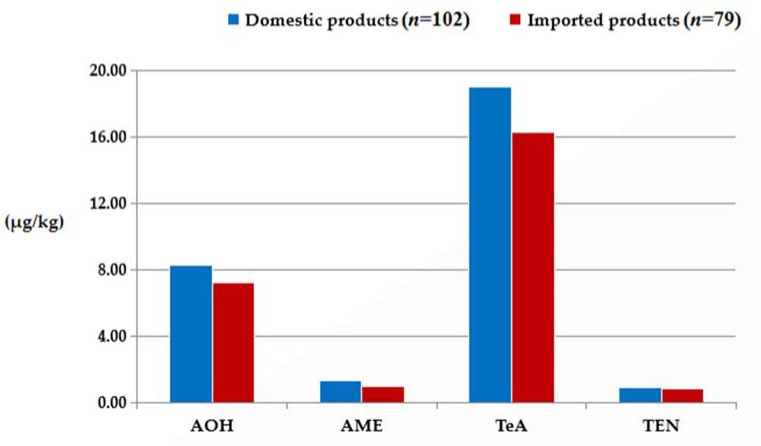
The average concentration distribution of the four ALTs in citrus-based products made in China and abroad.

**Table 1 toxins-15-00325-t001:** Natural occurrence of the four ALTs commercially available Chinese citrus-based products (*n* = 181) in 2021.

Mycotoxin	Range (μg/kg)	Average (μg/kg)	Median (μg/kg)	Frequency %
TeA	13.7–312.7	17.8	14.3	78.3 (142/181)
TEN	0.8–2.9	0.9	0.8	33.7 (61/181)
AOH	1.1–62.4	7.8	6.5	60.8 (110/181)
AME	0.7–13.4	1.2	0.8	70.7 (128/181)

**Table 2 toxins-15-00325-t002:** Contamination and co-contamination with the four ALTs in commercially available Chinese citrus-based products in 2021.

Contamination by	Toxin Combinations	Frequency % (*n*)
None (6.6%, 12/181)	-	6.6 (12/181)
One mycotoxin (10.0%, 18/181)	TeA	5.5 (10/181)
	AME	2.8 (5/181)
	AOH	1.7 (3/181)
Two mycotoxins (28.2%, 51/181)	AME-TeA	11.0 (20/181)
	AOH-TeA	7.2 (13/181)
	AOH-AME	3.9 (7/181)
	TeA-TEN	2.8 (5/181)
	AME-TEN	2.2 (4/181)
	AOH-TEN	1.1 (2/181)
Three mycotoxins (43.6%, 79/181)	AOH-AME-TeA	27.6 (50/181)
	AME-TeA-TEN	8.3 (15/181)
	AOH-AME-TEN	3.9 (7/181)
	AOH-TeA-TEN	3.9 (7/181)
Four mycotoxins (11.6%, 21/181)	TeA-TEN-AOH-AME	11.6 (21/181)

**Table 3 toxins-15-00325-t003:** Natural occurrence of the four ALTs in commercially available citrus-based samples made in China and abroad in 2021.

Region	Mycotoxin	Range (μg/kg)	Average (μg/kg)	Median (μg/kg)	Frequency %
Domestic products (*n* = 102)	AOH	1.1–62.4	8.3	6.6	60.8 (62/102)
	AME	0.7–14.4	1.3	0.8	73.5 (75/102)
	TeA	13.7–312.7	19.0	14.2	76.5 (78/102)
	TEN	0.8–2.9	0.9	0.8	31.4 (32/102)
Imported products (*n* = 79)	AOH	1.1–27.7	7.2	6.0	59.5 (47/79)
	AME	0.7–6.6	1.0	0.8	67.1 (53/79)
	TeA	13.7–53.2	16.3	14.4	81.0 (64/79)
	TEN	0.8–2.0	0.9	0.8	36.8 (29/79)

**Table 4 toxins-15-00325-t004:** Natural occurrence of the four ALTs in commercially available Chinese citrus-based samples in different physical forms in 2021.

Physical Form	Mycotoxin	Range (μg/kg)	Average (μg/kg)	Median (μg/kg)	Frequency %
Liquid products (*n* = 140)	AOH	1.1–62.4	7.5	6.0	60.0 (84/140)
	AME	0.7–6.6	1.0	0.8	70.0 (98/140)
	TeA	13.7–312.7	18.9	14.4	75.7 (106/140)
	TEN	0.8–2.9	0.9	0.8	36.4 (51/140)
Semi-solid products (*n* = 26)	AOH	1.4–12.6	5.4	4.8	53.8 (14/26)
	AME	0.7–2.6	1.0	0.9	65.4 (17/26)
	TeA	13.7–17.4	14.2	14.1	84.6 (22/26)
	TEN	0.8–0.8	0.8	0.8	30.7 (8/26)
Solid products (*n* = 15)	AOH	6.4–23.9	12.8	11.0	80.0 (12/15)
	AME	0.7–13.4	2.9	1.8	86.7 (13/15)
	TeA	13.8–18.2	14.7	14.5	93.3 (14/15)
	TEN	0.8–0.8	0.8	0.8	13.3 (2/15)

**Table 5 toxins-15-00325-t005:** Natural variety distribution of the four ALTs in different commercially available Chinese citrus-based products in 2021.

Kind	Mycotoxin	Range (μg/kg)	Average (μg/kg)	Median (μg/kg)	Frequency (%)
Oranges (*n* = 100)	AOH	1.1–27.7	7.4	6.5	60 (60/100)
	AME	0.7–6.6	1.0	0.8	75 (75/100)
	TeA	13.7–36.0	15.7	14.4	76 (76/100)
	TEN	0.8–0.9	0.8	0.8	32 (32/100)
Tangerines (*n* = 33)	AOH	1.4–23.9	9.3	6.9	63.6 (21/33)
	AME	0.7–13.4	2.0	1.0	75.8 (25/33)
	TeA	13.7–18.2	14.3	14.1	84.8 (28/33)
	TEN	0.8–0.8	0.8	0.8	15 (5/33)
Lemons (*n* = 31)	AOH	1.9–62.4	8.7	5.6	64.5 (20/31)
	AME	0.7–3.7	1.0	0.8	67.7 (21/31)
	TeA	13.7–62.5	17.9	14.2	80.6 (25/31)
	TEN	0.8–1.3	0.9	0.8	48.4 (15/31)
Pummelos and Grapefruits (*n* = 17)	AOH	1.1–9.5	5.0	5.0	52.9 (9/17)
	AME	0.7–1.2	0.9	0.8	41.2 (7/17)
	TeA	13.8–312.7	37.3	14.5	76.5 (13/17)
	TEN	0.8–2.9	1.2	0.8	52.9 (9/17)

## Data Availability

Data are contained within the article.

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
