# Peer review of "Natural Occurrence of Alternaria Toxins in Citrus-Based Products Collected from China in 2021"

_toxins, 2023, doi:10.3390/toxins15050325_

Round 1

Reviewer 1 Report

This manuscript describes the analysis of 181 citrus products from China for the four primary Alternaria toxins, alternariol, alternariol monomethyl ether, tentoxin and tenuazonic acid. At just 12 pages, 3 figures and 5 tables, it is a short manuscript as befits the relatively small study it describes.

The introduction is brief, to the point and is adequate to introduce the study. The sample size at 181 is adequate to provide statistical results and is well distributed between location of source, fruit type and fruit matrix (raw or processed). The results are presented as overall, domestic vs imported, differences between fruit species and differences between fruit consistency (liquid, semi-solid or solid).  The materials and methods is concise but detailed and well organized. The conclusions call attention to the fact that based on these results, exposure to these toxins due to consumption of fruit products originating in China requires further monitoring. Reasonable explanations for the trends shown in the results are offered – e.g. that there are higher concentrations in dried and concentrated liquids due to processing (see page 8). Consequences of this study are well outlined (Page 8 and 9, lines 288-292).

While the manuscript is well organized, there are many grammatical issues, incorrect use of words, etc., some of which are outlined below, that interrupt the flow of reading and need to be addressed.

Specific comments;

Page 2, lines 67-68: What is meant by “TEN was found in the lowest positive “rate””?

Lines 69-70: Incorrect grammar – incorrect use of “besides”. Try “In addition, no samples in this study contained AME and TEN at levels higher…”

Line 72: Reference 17 is a thesis, not Jiang et al.

Line 73: Could the authors offer a reason why these results are different from those of the references? Perhaps due to matrix difference – seeds, wheat?

Page 3: Lines 96-98: Simplify this sentence to make it more clear “Results are similar to those previously reported in which”?

Page 5, line 137: Grammatically incorrect – “samples from Chinese sources contained higher amounts of ALT’s than…..”

Line 141: The meaning of the sentence is unclear – do the authors mean that the processing methods are not as well controlled in China and therefore not as strict”?

Line 150-151: Can the authors give examples of what is meant by solid (dried fruit peel?), semi-solid (whole fruit?) and liquid (fruit juice?)

Page 6, lines 175-176 – This sentence make no sense – it is either moldy or not moldy. Are the authors insinuating that moldy fruit is included in processed products?

Page 7, lines 204, 229 – “In a word” is a phrase that should not be used in this context. Omit or use “Generally”

Page 8, line 242 – there is a word missing – “All of them are easily “contaminated” with ALT’s……”?

Line 247: “.. and should be given more attention”

Lines 256-259: Not sure if this is a valid reason for co-occurrence of at least two ALT’s. Most Alternaria produce more than one toxin in the field. Explain.

Lines 267-268: This is an awkward sentence. “Another cause could be that some moldy fruit would be included in the production process.”

Line 279: What is meant by “were easier”?

Page 9, line 317: What is meant by “waited”? -  “and further purified”?

Line 320: Omit “And”

Page 10, lines 344-345: Something is missing – “0.4 mg/kg and 20 mg/kg of what?” AME and TEN?

Reviewer 2 Report

The authors tracked the natural occurrence of Alternaria toxins in 2021 citrus-based products collected from China. Here are my suggestions and remarks. 

The title should be rearranged as suggested: Natural Occurrence of Alternaria Toxins in Citrus-based Products Collected in 2021from China

Abstract: specify what are citrus-based products (juice, concentrate, the fruit itself?)

Introduction: please add a paragraph and specify and describe the citrus-based products. 

Line 103: 

Line 141: you mention cans... is it canned juice or else? I am sure it is not a can, but a product in the can. Please clarify. 

In M&M you did not specify what are the samples, are they juice, or concentrate...? You mention solid samples, but do not mention which are they. Please specify the samples. 

Reviewer 3 Report

See the attachment please

Reviewer 4 Report

Dear Colleagues.
There are several questions, they are in the attached file
